# Prosperity or Real Estate Bubble? Exuberance Probability Index of Real Housing Prices in Chile

Byron J. Idrovo-Aguirre [1], Francisco J. Lozano [1] and Javier E. Contreras-Reyes [2,*]

1 Gerencia de Estudios, Cámara Chilena de la Construcción, Santiago 7560860, Chile; bidrovo@cchc.cl (B.J.I.-A.); flozano@cchc.cl (F.J.L.)
2 Instituto de Estadística, Facultad de Ciencias, Universidad de Valparaíso, Valparaíso 2360102, Chile
* Correspondence: jecontrr@uc.cl; Tel.: +56-(32)-250-8242

**Abstract:** In this paper, we approached the concept of real estate bubble, analyzing the risk its bursting could generate for the Chilean financial market. Specifically, we analyzed the relationship between real housing prices, the economic activity index, and mortgage interest rates denominated in inflation-linked units from 1994 to 2020. The analysis was based on a second order Markov switching model with the predetermined variables mentioned later, whose parameters were obtained through the expectation–maximization algorithm. Then, we built a probability index as early warning indicator for potential imbalances in the real estate price that could put financial market stability at risk. The indicator is important to evaluate economic policy calibrations in time. A main finding was that the real housing price had a non-linear relationship with economic activity and the mortgage interest rate. Therefore, the evolution of the real estate price has been consistent with fundamental macroeconomic variables, even under a high growth regime, with increases above 12% per year. About 92% of housing price variability derived from changing macrofinancial conditions, suggesting a low margin of speculative behavior.

**Keywords:** real housing prices; interest rate; policy decision making; Markov switching; expectation–maximization algorithm

## 1. Introduction

Real estate asset price analysis has been common in the international literature on housing markets and credit, especially after the dramatic effect the financial crisis had on the global economy in terms of productive activity, unemployment, and income (Agnello and Schuknecht 2011; Aguilera Alvial 2020; Balagyozyan et al. 2016; Cerutti et al. 2017; Gil-Alana et al. 2019; Helbling 2005). It is commonly believed that a sustained rise in home values is not based on favorable supply and demand conditions but indicates the existence of speculative bubbles. The main concern of the monetary authority then would be to implement actions that contribute to reducing the upward pressure on real estate prices to avoid an eventual burst of the bubble. Such a burst would mean a significant loss of wealth for families and investors since housing is usually among the most relevant assets in their savings and investment decisions.

Most studies on housing prices are based on the assumption that economic and financial conditions explain a high percentage of price dynamics (Goodhart and Hofmann 2008; Li and Li 2018). More specifically, considering the data usually available, economic conditions are often approximated with variables such as gross domestic product or disposable income, while financial conditions are approximated with the interest rate, either general or specific to mortgage credit. Some studies concluded that economic activity or income have a greater marginal effect on the appreciation of home values, compared to other explanatory variables such as construction costs or interest rates (Adams and Füss 2010). However, it is also generally understood that financial conditions should be more relevant for the evolution of housing prices (Helbling 2005) because the market indirectly intervenes in the transmission of monetary policy (Elbourne 2008). In this sense, interest rates turn out to be good predictors for both

the movement of housing prices (Anundsen and Jansen 2013; Chen et al. 2014) and boom and bust cycles experienced by the real estate sector (Agnello and Schuknecht 2011; Cerutti et al. 2017; Duca et al. 2021; Martínez and Oda 2021).

Traditionally, real estate bubbles have been identified through linear models that contrast the alignment of the housing price with its economic fundamentals. If permanent divergences were found between property values and their explanatory variables, the hypothesis on speculative behavior in the housing market would be accepted. Unit root and cointegration tests have been the main tools for testing this hypothesis. However, several authors identified limitations to these methodologies. Evans (1991) found that traditional tests are not useful in detecting bubbles that collapse on a recurring basis. Flood and Hodrick (1990) referred to the difficulty of estimating a fundamental price (based on fundamentals) with which to compare the observed price, while Gürkaynak (2008) considered the complication of differentiating between bubble and regime change in the fundamentals. Kim and Min (2011) highlighted the specification errors that arise when proposing a regression model to estimate the presence of bubbles. In response, some authors proposed considering non-linear relationships between housing prices and their fundamentals through the estimation of regime change models. This way, it would be possible to characterize the different states of expansion and collapse typical of financial markets (Balcombe and Fraser 2017; Elhafidi and Ouchen 2016). van Norden and Vigfusson (1996) showed that tests to identify bubbles based on regime switching are superior to those based on unit roots.

The application of models with regime changes to analyze real estate sector cycles has become more frequent, especially after the international financial crisis. Cerón and Suárez (2006), using a panel of 14 OECD countries, found two regimes for the dynamics of housing prices according to their volatility and a direct relationship with economic and financial conditions. Fontana and Corradin (2013) identified three regimes for rising property prices in Europe: high, medium, and low. Chowdhury and Maclennan (2015) distinguished two states for the United Kingdom: one with high growth and low volatility, the other with low growth and high volatility. Many studies focused on the United States, showing the existence of two regimes that govern the dynamics of the housing sector. Chen et al. (2014) found a state of high volatility related to boom-and-bust episodes and another of low volatility. Kim and Chung (2014) evaluated the relationship between price and income, finding a long-term stability regime, as predicted under the present value method, and another non-linear relationship regime. Balagyozyan et al. (2016) identified high and low stages for the growth of housing prices and showed that these cycles lead the economic cycle Duca et al. (2021). Likewise, for the United States, other studies found three regimes in the evolution of housing prices. Nneji et al. (2013) found states of boom, stability, and crash in residential market dynamics, while Prüser and Schmidt (2021) identified national boom regimes, national bust regimes, and local bust regimes, highlighting each region's characteristics.

There are also applications of regime change models in estimating cycles in the real estate sector in other countries. Among others, Feng and Li (2011) used a model of changes in the housing price regime in Beijing (China), detecting bubble episodes and confirming the correlation with their macrofinancial determinants. Kim and Min (2011) observed two states in South Korean price dynamics according to the level of volatility. Simo-Kengne et al. (2013) estimated an autoregressive vector with regime changes for the South African housing market, distinguishing expansive (bull) and contractive (bear) states. Savva (2015) selected an optimal two-regime structure (boom and recession) for the evolution of housing prices in Cyprus. Espinosa and Sanin (2016) found high and low volatility regimes in the Colombian housing market, the former being the least common. Additionally, other studies selected an optimal number of regimens out of three: Sethapramote et al. (2019) analyzed housing price dynamics in Chinese provinces, finding fast, normal, or low states in line with price growth. Rangel and Ng (2017) used a three-regime structure for the Singaporean residential market: boom, steady state, and crash.

Almost all these studies aimed to establish a non-linear structure for the housing price and to estimate the effect of economic and financial conditions on real estate cycles (Duca et al. 2021; Martínez and Oda 2021). In general, the most recurrent explanatory

variables in this type of estimations were aspects of economic activity (gross domestic product, activity rate, unemployment, income, among others) and indicators related to financial conditions (interest rates, credit growth, and lending standards). They found marginal effects on price dynamics that are consistent with economic theory. Economic shocks on variables positively impact the sector cycle, while shocks on financial conditions impact dynamics negatively. For example, higher disposable income causes expansions in real estate demand, while a rise in the interest rate generates a contraction.

Various stylized facts appear in the literature, mainly related to the asymmetries inherent to the business cycle. A first type of asymmetry is related to the duration and magnitude of boom-and-bust periods, the latter being more frequent and long-lasting (Cerón and Suárez 2006). On the other hand, there is also an asymmetry of the effect of economic and financial conditions on price evolution. In most cases it was shown that shocks generate more impact in expansive periods, with this marginal effect being less relevant or even insignificant during stability or recession periods (Chowdhury and Maclennan 2015; Espinosa and Sanin 2016; Nneji et al. 2013; Rangel and Ng 2017; Savva 2015). Finally, asymmetries were found in the real estate cycles at the geographic and administrative level, that is, each state, region, or province may exhibit different regimes in terms of timing, intensity, and duration (Fontana and Corradin 2013; Prüser and Schmidt 2021; Sethapramote et al. 2019).

Analyzing the evolution of housing prices is essential to assess and anticipate financial stability risks. One of these risks is the formation of real estate bubbles, with potential negative impacts on financial markets and well-being. Therefore, evaluations of asset price growth relative to the performance of fundamental variables such as income and mortgage interest rate (Alegría and Bravo 2016) have become more relevant in policy decision making, especially after the 2008 global financial subprime crisis (Guo et al. 2011). In this sense, it is imperative to create an early warning indicator allowing a glimpse at each moment of the probabilities when the market price could enter different regimes. The indicator could help evaluate calibrations to economic policy (Idrovo-Aguirre and Contreras-Reyes 2021a).

For this purpose, we have developed a housing price exuberance probability index. It enables visualizing on a monthly basis the probability of housing prices entering an expansionary growth cycle and highlights the extent to which this exuberance is consistent with fundamental macroeconomic variables. The indicator was prepared based on the methodology used by Johnson (2001) for the creation of an artificial monetary policy modification index in Chile. However, our index was constructed by identifying asymmetries or anomalies in the distribution of annual housing price growth, conditional upon macroeconomic factors related to market supply and demand. These asymmetries represent the different "states" of the real housing price growth rate, and these states can be observed with certain probability. Crucially, the Central Bank's Financial Stability Report mentioned the term housing price exuberance, based on (Martinez-García and Grossman 2020). This term refers to a housing price growth deemed unsustainable relative to its fundamental variables such as the economic cycle and mortgage interest rates. Therefore, the housing price exuberance probability index accounts for the likelihood of a fundamental misalignment of real estate prices that could jeopardize financial market stability in Chile. This index might be useful as an additional signal employed by the monetary authority in conducting its mandate in order to evaluate potential risks.

Studies that focus on early warning indicators related to real estate activity in Chile did not previously exist. Those that mentioned the Chilean housing market were based on linear time series estimates to contrast the economic factors that affect the evolution of housing prices, particularly by estimating Vector Error Correction models (Aguilera Alvial 2020; Gil-Alana et al. 2019), Idrovo and Lennon (2012, 2013); (Silva and Vio 2015). Although the economic literature on the local housing market contains multiple applications to understand the sector's recent evolution, there is little effort, even negligence, to generate forecasts and identify leading indicators that could serve as guides for policymakers. This work's innovations relate to a non-linear estimation of housing price determinants and the calculation of a probability index to evaluate risks derived from real estate activity. Furthermore, analyzing the case of Chile is relevant for two reasons. First, the country has one of the most developed housing and credit markets within the region, thus offering

long and trusted time series for housing prices, credit growth, and interest rates. Second, the country did not suffer a significant bubble burst for the last two decades, even though housing prices have been rising for a long time.

The probabilities were calculated based on the estimation of regime switching models, originally proposed by Hamilton (1990) and extended upon by Kim and Nelson (1999)[1]. In our case, the application of these models supposed transitory but recurrent instabilities in the intertemporal dynamics of the housing price relative to macroeconomic determinants. In other words, these models assume that the jumps or inflection points observed in the real housing price growth rate can be characterized by mean and conditional variance parameters that change recurrently from one subperiod (or regime) to another. Our model allows simulating the housing price behavior conditioned on two variables linked to supply and demand. These variables are predetermined loan interest rate in inflation-linked units for housing (Alegría and Bravo 2016) and the annual growth of monthly economic activity index (Firinguetti and Rubio 2003). Both variables are published by the Central Bank of Chile (Banco Central de Chile, https://www.bcentral.cl/, accessed on 28 May 2021). Intuitively, the mortgage rate captures the situation of the financial conditions for access to housing, i.e., whether these conditions are favorable (low interest rate) or restrictive (high interest rate). Meanwhile, economic activity index dynamics partially reflect the evolution of average consumer incomes and those of investors in this market.

The contribution of our paper to the literature is threefold. First, the existing linear models in the local economic literature require multiple variables (Aguilera Alvial 2020) or fractionally integrated processes (Gil-Alana et al. 2019) to achieve an adjustment similar to that derived from our non-linear model with two elemental determinants: economic activity index and mortgage interest rate. In particular, Vector Error Correction models (VEC) are the most commonly used to account for the cointegration relationship of the price with its fundamental variables. Therefore, the advantage of our Markov chain model with predetermined variables over classical linear regression or time series models is that it allows telling a story about the key determinants of price non-linearity. Second, from the results of the switching model for the estimation of the housing price conditional on economic activity index data and mortgage loan interest rates, it followed that 92% of the variability of the price was explained by changes in macrofinancial conditions, suggesting a low margin of speculative behaviors. In this sense, the episodes of high housing prices responded more to a scenario of prosperity rather than speculative bubbles in the real estate market. Third, and based on this last result, we created a systematic index of real housing price exuberance in Chile that could be considered for monetary policy decisions. Its evolution highlights the crucial role economic policy plays in the dynamics of the real estate price versus market speculations that imply a risk alert for financial stability. In fact, housing price dynamics is helpful for monetary policy decisions and a crucial variable that affects inflation (housing prices are transformed to rental market) and financial stability (houses and investors with high mortgage credit debt).

This work is organized as follows. Section 2 shows a formal development of the regime change models and the parameter estimation method. Section 3 presents a descriptive analysis of the database. In particular, it analyzes the evolution of the real property value index in the Metropolitan Region and breakdowns by type of house and geographical area. Section 4 presents the main results of the model estimates and Section 5 contains the main conclusions.

## 2. Methodology

### 2.1. Regime Change Model with Predetermined Variables

Consider the following regime change (switching) model that relates the annual growth of housing prices with its fundamental variables, measured in real terms:

$$p_t = \mathbf{x}_{t-1}\boldsymbol{\beta}_t(s) + \varepsilon_t, \tag{1}$$

$$\varepsilon_t \sim^{iid} N(0, \sigma_{s_t}^2), \tag{2}$$

with

$$\boldsymbol{\beta}_t(s) = \sum_{m=1}^{M} \boldsymbol{\beta}_m S_t(m), \tag{3}$$

$$\sigma_{s_t}^2 = \sum_{m=1}^{M} \sigma_m^2 S_t(m), \tag{4}$$

$$S_t(m) = \begin{cases} 1, & \text{if } S_t = m \quad (m\text{th state});\\ 0, & \text{otherwise.} \end{cases} \tag{5}$$

where, $p_t$ is the annual growth in housing prices at each instant of time ($t$). The set of variables $\mathbf{x}_{t-1} = [1, r_{t-1}, y_{t-1}]$ is a time series row vector of dimension $(1 \times 3)$ that contains, in addition to the regression constant, the housing loan interest rate in inflation-linked units ($r_t$) and the annual economic activity index variation ($y_t$). Both variables ($r_t, y_t$) are related to supply and demand (Lozano 2015b). The lagged or predetermined structure in the model ($\mathbf{x}_{t-1}$), in addition to protection from potential endogeneity biases, considers adjustment costs implicit in the purchase or investment of real estate to macroeconomic conditions on which agents decide (timing of approval of mortgage loans and notarial procedures, for example).

On the other hand, $\boldsymbol{\beta}_t(s) = [\mu_t(s), \delta_t(s), \gamma_t(s)]^\top$ is a column vector of dimension $(3 \times 1)$, which contains the variable coefficients of the model: (i) the intercept of the regression ($\mu_t(s)$), (ii) the parameter of the real property value annual growth/real mortgage interest rate relationship ($\delta_t(s)$), and (iii) the coefficient that relates housing price growth to economic activity ($\gamma_t(s)$). Unlike in classic linear regression models, here the model parameters vary according to the original state $S_t(m)$ of annual housing price growth, with $m = \{1, ..., M\}$, $M \in \mathbb{N}$, and $M$ the maximum number of possible states of the annual variation. Finally, $\varepsilon_t$ is the error term of the model. It is assumed that it follows a normal distribution with zero mean and variance conditional on the $m$-th state $S_t$ ($\sigma_{s_t}^2$). So, if $S_t$ corresponds to a particular state $m \in M$, the dichotomous indicator $S_t(m)$ takes value 1 in this case and value 0 in any other. Indeed, the parametrization of the model occurs when $S_t = m$ is such that $\boldsymbol{\beta}_t(s) = \boldsymbol{\beta}_m$ and $\sigma_{s_t}^2 = \sigma_m^2$.

Moreover, the model assumes that states ($S_t$) can be observed with certain probability. In turn, this probability depends on the state observed in the immediately preceding period ($S_{t-1}$) and on the set of information available ($\boldsymbol{\psi}_t$) up to time $t$, that is, $\boldsymbol{\psi}_t \equiv \{p_t, \mathbf{x}_{t-1} \boldsymbol{\psi}_{t-1}\}$. Therefore, discrete variable $S_t$—which defines the state of nature that governs the housing price behavior—is assumed to follow a Markov chain process, known in the literature as the Markov Switching Model (Guidolin and Timmermann 2008; Guo et al. 2011; Johnson 2001; Kim and Nelson 1999; Kim et al. 2008; Pelletier 2006).

Formally, these probabilities between states' transitions (or subperiods) are synthesized by matrix $\mathbf{P} = [p_{ij}]$, $i, j = 1, \dots, M$. Each element

$$p_{ij} \equiv \mathbb{P}(S_t = i | S_{t-1} = j, \boldsymbol{\psi}_t) \tag{6}$$

represents the probability that the annual growth rate of housing prices derives from a regime $i$ ($S_t = i$) since it passed through a state $j$ ($S_{t-1} = j$). For example, if state $S_t = 1$ represents a regime of "high" housing price growth and state $S_{t-1} = 2$ corresponds to a regime of "moderate" growth, then

$$p_{12} \equiv \mathbb{P}(S_t = 1 | S_{t-1} = 2, \boldsymbol{\psi}_t) \tag{7}$$

displays the probability that the growth rate of real housing prices will rise from "moderate" to "high".

Another point that emerged from this model is that, based on Bayes' theorem, it is possible to identify the conditional probability that the housing price passes through some specific growth regime at each instant of time. This measure will be relevant for the construction of the exuberance probability index. Formally, these conditional probabilities are:

$$\mathbb{P}(S_t = i | p_t, \mathbf{x}_{t-1}, \boldsymbol{\psi}_{t-1},) \quad = \quad \sum_{j=1}^{M} p_{ij} \mathbb{P}(S_{t-1} = i | \boldsymbol{\psi}_{t-1}) \tag{8}$$

where $\mathbb{P}(S_t = i | p_t, \mathbf{x}_{t-1}, \boldsymbol{\psi}_{t-1})$ is the conditional probability that the observed housing price passes through some specific growth regime $i$ at a given moment of time $t$. This way, it is possible to identify not only the original state in which the real housing price is but also to evaluate ex post the real estate market's price cycles and the relationship with its fundamental macroeconomic variables. In this sense, the significance of the macroeconomic determinants for each growth regime, synthesized in a probability index, allowed an appropriate assessment of the risks to financial stability.

### 2.2. Estimation Strategy: Expectation–Maximization Algorithm

The model parameters are estimated using the expectation–maximization algorithm (Contreras-Reyes et al. 2014; Dempster et al. 1977; Hamilton 1990). Let $\boldsymbol{\theta} = [\boldsymbol{\theta}_1^\top, \boldsymbol{\theta}_2^\top]^\top$ be the set of Markov switching model parameters so that:

$$\boldsymbol{\theta}_1 \quad = \quad [\boldsymbol{\beta}_1, \ldots, \boldsymbol{\beta}_M, \sigma_1^2, \ldots, \sigma_M^2]^\top, \tag{9}$$

$$\boldsymbol{\theta}_2 \quad = \quad [p_{11} \ldots p_{M-1 1}, \ldots, p_{1M}, \ldots, p_{M-1 M}]^\top \subset vec(\mathbf{P}), \tag{10}$$

where $vec(\mathbf{P})$ is the vectorization of probability matrix $\mathbf{P}$ transition. Therefore, the function to be maximized is:

$$\mathscr{Q}(\boldsymbol{\theta}; \tilde{\mathbf{p}}_T, \boldsymbol{\theta}^{k-1}) \quad = \quad \int_{\tilde{\mathbf{S}}_T} \sum_{t=1}^{T} \log\{f(p_t | S_t, S_{t-1}; \boldsymbol{\theta}_1^k) \mathbb{P}(S_t, S_{t-1}; \boldsymbol{\theta}_2^k)\} \mathbb{P}(\tilde{\mathbf{p}}_T, \tilde{\mathbf{S}}_T; \boldsymbol{\theta}^{k-1}), \tag{11}$$

s.t.

$$f(p_t | S_t, S_{t-1}; \boldsymbol{\theta}^k) \quad = \quad \frac{1}{\sqrt{2\pi\sigma_{s_t}^2}} e^{-\frac{1}{2}\frac{[p_t - \mathbf{x}_{t-1}\boldsymbol{\beta}_t(s)]^2}{\sigma_{s_t}^2}}, \tag{12}$$

$$\sum_{i=1}^{M} p_{ij} \quad = \quad 1, \tag{13}$$

$$p_{ij} \quad \geq \quad 0, \tag{14}$$

where $\tilde{\mathbf{p}}_T = [p_1, \ldots, p_T]^\top$ is the time series of the real property value index growth rate in the Metropolitan Region, a variable observable from January 1995 to December 2020; $\tilde{\mathbf{S}}_T = [1, \ldots, S_T]^\top$ is the state variable, not observable but probabilistically identified, and $\boldsymbol{\theta}^k$ is the parameter vector resulting from the $k$-th iteration of this maximization method.

From the necessary first order conditions, the result of the parameters that define the structure of the expectation–maximization algorithm was obtained:

$$\hat{\boldsymbol{\beta}}_t^{(k)}(s) \quad = \quad \left[\sum_{t=1}^{T} \mathbf{x}_{t-1}^\top \mathbf{x}_{t-1} \mathbb{P}(S_t | \tilde{\mathbf{p}}_T, \boldsymbol{\theta}^{(k-1)})\right]^{-1} \sum_{t=1}^{T} \mathbf{x}_{t-1}^\top p_t \mathbb{P}(S_t | \tilde{\mathbf{p}}_T, \boldsymbol{\theta}^{(k-1)}), \tag{15}$$

$$\hat{\sigma}_t^{2(k)}(s) \quad = \quad \frac{\sum_{t=1}^{T} [p_t - \mathbf{x}_{t-1}\boldsymbol{\beta}_t(s)]^2 \mathbb{P}(S_t | \tilde{\mathbf{p}}_T, \boldsymbol{\theta}^{(k-1)})}{\sum_{t=1}^{T} \mathbb{P}(S_t | \tilde{\mathbf{p}}_T, \boldsymbol{\theta}^{(k-1)})}, \tag{16}$$

Finally, as many iterations as necessary were carried out until $\boldsymbol{\theta}^{(k)} = \boldsymbol{\theta}^{(k-1)}$. For this purpose the following convergence (stop-rule) criterion was used:

$$|(\boldsymbol{\theta}^{(k)} - \boldsymbol{\theta}^{(k-1)})^\top (\boldsymbol{\theta}^{(k)} - \boldsymbol{\theta}^{(k-1)})| \quad < \quad \epsilon, \tag{17}$$

with $\epsilon > 0$. Lastly, the variance-covariance matrix of the vector of coefficients $\hat{\boldsymbol{\beta}}_t^{(k)}(s)$ was

obtained from the lower Cramer–Rao bound,

$$\hat{V}(\hat{\boldsymbol{\beta}}_t^{(k)}(s)) \;=\; \hat{\sigma}_t^{2(k)}(s)\left[\sum_{t=1}^{T} \mathbf{x}_{t-1}^{\top}\mathbf{x}_{t-1}\mathbb{P}(S_t|\tilde{\mathbf{p}}_T,\boldsymbol{\theta}^{(k-1)})\right]^{-1}. \tag{18}$$

with these parameters it was possible to define a confidence interval for the estimated coefficients of the predetermined real property value index growth. Formally, we have:

$$\mathbb{P}\left(\hat{\boldsymbol{\xi}}_i - t_{1-\frac{\alpha}{2}}\sqrt{\hat{\boldsymbol{\Sigma}}_{ii}} \le [\boldsymbol{\beta}_t^{(k)}(s)]_i \le \hat{\boldsymbol{\xi}}_i + t_{1-\frac{\alpha}{2}}\sqrt{\hat{\boldsymbol{\Sigma}}_{ii}}\right) \;=\; 1-\alpha, \tag{19}$$

where $\hat{\boldsymbol{\xi}}_i = [\hat{\boldsymbol{\beta}}_t^{(k)}(s)]_i$ is the $i$th coefficient parameter $[\hat{\mu}_t^{(k)}(s), \hat{\delta}_t^{(k)}(s), \hat{\gamma}_t^{(k)}(s)]^{\top}$, $t_{1-\frac{\alpha}{2}}$ is the critical value obtained through distribution tables $t$ with a level of significance $\alpha$, and $\hat{\boldsymbol{\Sigma}}_{ii} = [\hat{V}(\hat{\boldsymbol{\beta}}_t^{(k)}(s))]_{i,i}$ is the $i$th element of the main diagonal of the variance-covariance coefficient matrix. Its square root is the standard error of the $i$th coefficient.

The expectation–maximization algorithm for estimating the parameters was programmed in MATLAB Code (Matlab 2019) with a maximum of 500 iterations ($k_{max} = 500$).

## 3. Data

In this section we present retrospect data used for estimating the housing price growth model and sketch the construction of the probability index. First, we will examine the evolution of the real property value index of the Metropolitan Region and its dissections, both by type of unit (houses and apartments) and by geographic zones. Real property value index figures and their breakdowns are published by the Chilean Chamber of Construction (Cámara Chilena de la Construcción) on a monthly basis (Idrovo and Lennon 2011). Later, we will describe economic activity index behavior and the mortgage interest rate, whose data source is the Central Bank of Chile. Both variables are presumed to be macroeconomic determinants of housing prices, due to their relationship with real estate market supply and demand.

The preparation of the real property value index and its breakdowns were based on the hedonic price methodology (Idrovo and Lennon 2011; Lozano 2015a). The interest in studying real property value index dynamics with respect to other price indicators in Chile has a long record, with monthly observations from January 1994 to December 2020 (324 records). Therefore, the historical path of this indicator includes four well-marked periods of Chile's economic cycle: (i) The impact of the 1999 Asian crisis (Contreras-Reyes and Idrovo-Aguirre 2020; Idrovo-Aguirre and Contreras-Reyes 2019; Ramírez-Parietti et al. 2021), (ii) the effect of the 2008–2009 financial subprime crisis (Contreras-Reyes and Idrovo-Aguirre 2020; Idrovo-Aguirre and Contreras-Reyes 2019), (iii) the mining investment boom during 2014–2015 (Idrovo-Aguirre and Contreras-Reyes 2021b), and (iv) the start of the COVID-19 health crisis in March 2020 (Mena et al. 2021).

In addition, the annual real property value index growth was significantly correlated with the 12-month variation in the quarterly property value index in the Metropolitan Region, prepared by the Central Bank based on the stratification methodology (BCCh 2014, 2019). The correlation coefficient between both variables is $\rho = 0.56$ (with a $p$-value = 0.00). Crucially, the property value index in the Metropolitan Region was measured from the first quarter of 2002 (76 observations up to the fourth quarter of 2020) and publication had a lag of six months. Therefore, the real property value index of the Chilean Chamber of Construction, being published monthly, could be interpreted as a naturally advanced indicator of the property value index in the Metropolitan Region. In addition, the real property value index is calculated with sales promises, while the Central Bank's property value index is calculated with deeds. Thus, there is usually an average of 6 to 12 months between promise and deed, sometimes 18 or more. Moreover, the correlation between the real property value index and Central Bank's property value index is $\rho = 0.46$ with a significant 95% confidence. Likewise, high positive correlation is observed between the property value index in the Metropolitan Region and the property value index ($\rho = 0.94$).

Therefore, the dynamics of the housing price in the Metropolitan Region strongly reflect national real estate price behavior.

In retrospect, the real property value index grew annually with an upward trend—in terms of its average and dispersion—during the monthly period from January 1995 to December 2020 (Figure 1). Therefore, the historical pattern of real housing prices revealed an increase not only in the average market value of real estate assets owned by households and investors but also in their volatility. This anomalous behavior also implied greater difficulties in market access for vulnerable families who require a housing solution, especially in a context of prolonged high unemployment because of the health crisis caused by COVID-19. Thus, in quantitative terms, the average and dispersion of annual real property value index growth seemed to increase substantially as of 2008. Subperiods selected were based on before and after the international financial subprime crisis. In particular, the average 12-month expansion observed since 2008 (5.6%) was 20 times higher than that registered from 1995 to 2007 (prior to the subprime crisis). Likewise, its volatility has more than doubled in the same period, probably accounting for rising uncertainty.

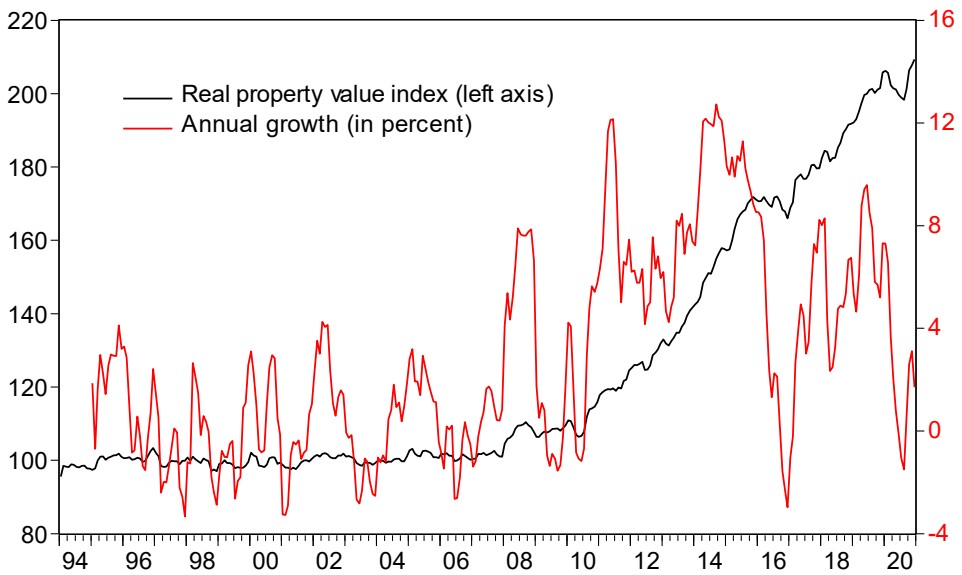

**Figure 1.** Evolution of the real property value index in Chile's Metropolitan Region and its annual growth rate (1994–2020).

A pattern similar to the aggregate real property value index trend was found for apartment prices, consistent with the greater variance and positive trend in the prices of Central Santiago and in the northeast of the Metropolitan Region (Figure 2). In statistical terms, some price movements were observed disaggregated by type of housing and area. For example, the annual growth rate for housing showed significant correlations of 0.94 and 0.59 for apartment and house prices, respectively. Likewise, the correlation of apartment price variation in Central Santiago and the northeast Metropolitan Region with the general growth of the real property value index was 0.62 and 0.85, respectively.

Housing prices rose at a relatively constant average rate over time, without any marked trend in historical evolution. However, the volatility of the real property value index in the Metropolitan Region's northeast and south showed behavior similar to that deduced from the valuations of apartments in general. Therefore, although price growth rates with a positive trend were not observed in all types of housing and geographical areas, the growing dispersion around the price of the real estate market seems to have been a common element in the interior of the Metropolitan Region during the last decade.

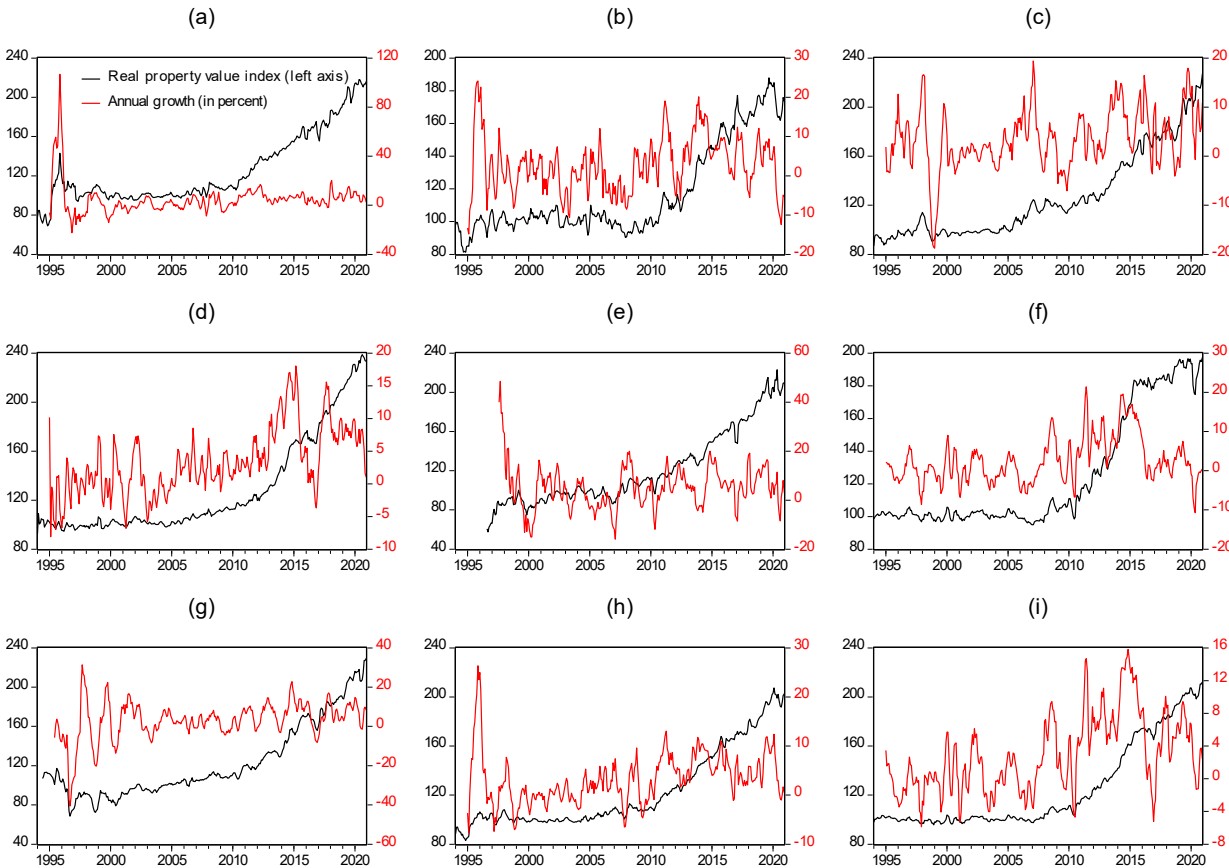

**Figure 2.** Monthly real property value index and its annual growth rate (1994–2020), disaggregated by type of housing (house and apartment) and according to geographic area: (**a**) house prices in Metropolitan Region northwest, (**b**) in the northeast, (**c**) in the south; (**d**) apartment prices in Central Santiago, (**e**) in Metropolitan Region northeast, (**f**) in the northwest, (**g**) the south; (**h**) house prices in the Metropolitan Region; (**i**) apartment prices in the Metropolitan Region.

This apparent instability in the mean and dispersion of the annual growth rate of the real housing price could indicate the existence of multipolarities in its evolution. In other words, the real property value index could be experiencing changes of states or regimes that govern its trajectory over time. Therefore, contrasting the presence of asymmetries in the growth rate distribution and identifying the contribution or implication of its macroeconomic determinants is inexorably necessary to avoid misinterpretations of real estate bubbles during the boom of this asset market.

For this purpose, we considered two fundamental macroeconomic variables of the real estate market to explain real property value index dynamics analyzed in Johnson (2001). One is the mortgage interest rate in inflation-linked units for housing (or real mortgage interest rate), which contains information on the financial conditions for access to housing. The other is economic activity index. The economic activity index measures the average income behavior of agents' that participate in this market. Table 1 summarizes the definition and source of these variables.

Table 2 presents a statistical summary of the variables considered in the estimation of the real property value index growth model and the construction of the probability index. In particular, the Jarque–Bera statistics (Jarque and Bera 1987) reject the hypothesis of normal real property value index growth, confirming distribution asymmetries.

**Table 1.** Definition of variables considered in this study.

| Variable | Notation | Definition | Source |
|---|---|---|---|
| Real property value index in Santiago, Chile | $p_t$ | Measures the evolution of real housing price index methodology and is based on hedonic price estimation techniques. | Idrovo and Lennon (2011) |
| Economic activity index | $y_t$ | Variable that reflects the evolution of Chile's economic cycle and the life cycle income of consumers and investors in the real estate market. | BCCh (2006); Escandón et al. (2005) |
| Mortgage interest rates denominated in inflation-linked units | $r_t$ | Variable that includes mortgage credit costs and financial conditions for accessing housing in Chile. | BCCh (2021) |

**Table 2.** Statistical summary of the variables to be considered in the regime change model and creation of the probability index (1994–2020).

| | Real Property Value Index (Annual Variation, in %) | Credit Rate Mortgage (in %) | Economic Activity Index (Annual Variation, in %) |
|---|---|---|---|
| Mean | 2.92 | 5.65 | 4.09 |
| Median | 2.01 | 4.53 | 4.20 |
| Maximum | 12.72 | 13.54 | 23.41 |
| Minimum | −3.34 | 1.99 | −15.52 |
| Stand. Dev. | 3.99 | 2.57 | 4.39 |
| Asymmetric coef. | 0.59 | 0.88 | −0.45 |
| Kurtosis | 2.41 | 2.43 | 7.25 |
| Jarque-Bera (*p*-value) | 0.00 | 0.00 | 0.00 |

Figure 3 shows the relationship of the real property value index, real mortgage rate, and economic activity index variables. At least two conglomerates of price volatility emerged from the interest rate-housing price relationship. This heterogeneity indicates a relation between the non-linear behavior of real property value index and the financial conditions for access to housing credit. In general, decreases in the real rate of mortgage loans tended to coincide with greater dispersion of real property value index. Likewise, the correlation between both variables was negative (−0.47) and significant at 95% confidence.

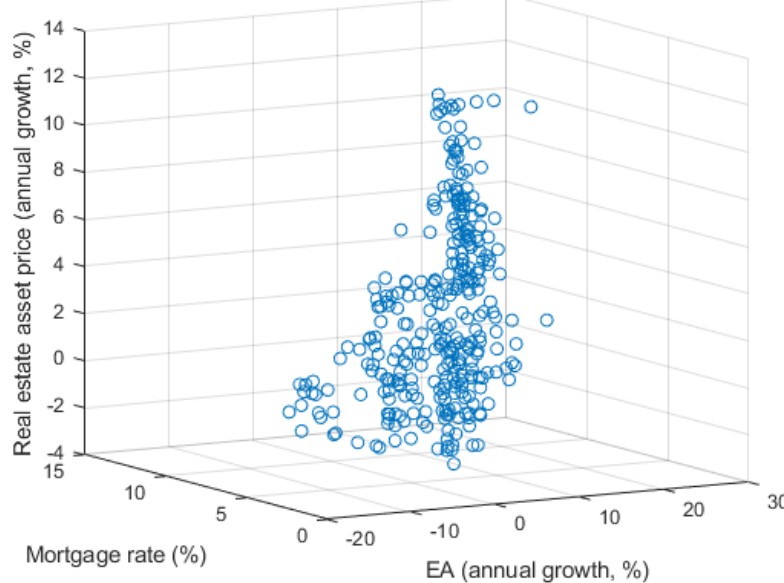

**Figure 3.** Real property value index, credit rate in inflation-linked units for housing and economic activity index (1995–2020).

From the economic activity index perspective, a somewhat diffuse relationship between its annual growth rate and real property value index fluctuations was observed. However, Figure 3 shows that in the episodes in which economic activity exceeded a 5% growth threshold and coincidentally the real mortgage interest rate was relatively low at less than 4%, housing prices grew significantly, around 12%. In this case, the macroeconomic conditions justified a scenario of prosperity for the real estate market since housing price dynamics seemed in line with the performance of fundamental variables.

In the following sections, we evaluate if the asymmetry or anomaly of the real property value index growth rate is consistent with the behavior of its macroeconomic determinants, this time considering the presence of regime changes in its distribution (switching model).

## 4. Results

According to the information criteria, a regime change model with three states of nature ($M = 3$) is appropriate to obtain a good fit between the annual growth of the real estate price, the annualized rate for housing loans in inflation-linked units, and the annual economic activity index variation. Table 3 shows that the estimation of a second order Markov switching model (three alternative states) yields the lowest critical value for each of the three information criteria considered here.

**Table 3.** Information criteria for fitting the switching model with predetermined variables, according to alternative states of nature.

| Information Criteria | Alternative States or Regimens | | | |
|---|---|---|---|---|
| | $M = 2$ | $M = 3$ | $M = 4$ | $M = 5$ |
| Akaike | 4.41 | 4.05 | 5.10 | 5.10 |
| Hannan–Quinn | 4.46 | 4.14 | 5.19 | 5.22 |
| Schwarz | 4.53 | 4.27 | 5.33 | 5.39 |

Figure 4 shows a rapid convergence of the estimated parameters of the regime change model, with three alternative states for the annual real property value index growth rate (second order Markov switching Model). The estimate is conditional on two predetermined economic variables: annual economic activity index growth and annualized mortgage interest rate in inflation-linked units. Thus, the sequences of the parameters calculated by the expectation–maximization algorithm converged from iteration 92 ($k_0 = 92$), also considering a maximum of 500 iterations as the limit ($k_{max} = 500$). The convergence rates in Figure 4 were robust to different initial values or departures of the model parameters. Table 4 summarizes the values to which the model parameters converged based on the expectation–maximization algorithm. These values allow characterizing the distribution of the annual real property value index growth rate as a mixed distribution with three alternative regimes and conditional on their default variables or macroeconomic fundamentals.

Table 4 also shows the average real property value index growth rate (or long-term growth) is 3.3% per year, similar to the average annual inflation (3.2%) from January 2000 to December 2020. Likewise, the coefficients of the regression model that define this average growth rate were significant at 95% of confidence level, as can be concluded based on the magnitude of their respective standard errors. Therefore, the average real property value index growth rate was explained both by the average income of consumers and investors (included in the economic activity index dynamics) and by the financial conditions faced by these agents (reflected in the evolution of the real mortgage interest rate).

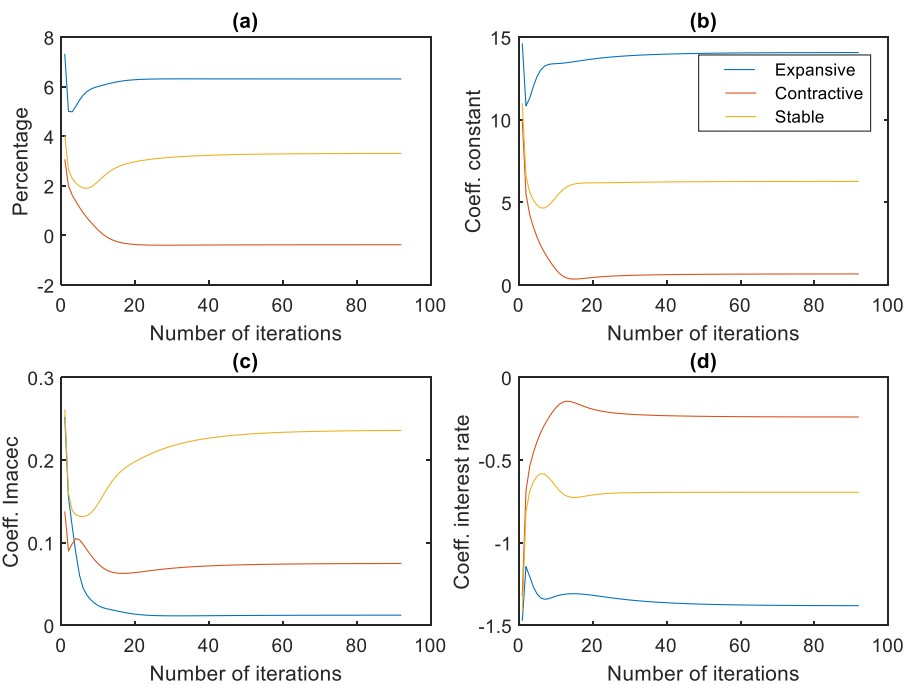

**Figure 4.** Evolution of the convergence rates of the EM algorithm. (**a**) Annual real property value index growth (in %); (**b**) Coefficient of the regression constant; (**c**) Coefficient that accompanies annual economic activity index growth; (**d**) Coefficient that accompanies the mortgage interest rate.

**Table 4.** Convergence rate for different annual growth regimes of the real property value index. The values in parentheses correspond to the standard error of the coefficients that accompany economic activity index and loan interest rates, respectively.

| Description | Parameters | Expansive $(S_t = 1)$ | Medium $(S_t = 2)$ | Contractive $(S_t = 3)$ |
|---|---|---|---|---|
| Conditional growth (in %) | $\bar{p}_t = \bar{\mathbf{x}}_{t-1} \hat{\boldsymbol{\beta}}_t^{(k_0)}(s)$ | 6.314 | 3.310 | −0.373 |
| Conditional Stand. Dev. (in %) | $\hat{\sigma}_t^{2(k_0)}(s) \in \hat{\boldsymbol{\theta}}^{(k_0)}$ | 2.077 | 1.738 | 1.353 |
| Long-term probability (in %) | $\mathrm{E}[\mathbb{P}(S_t|\hat{\boldsymbol{\theta}}^{(k_0)})]$ | 0.358 | 0.338 | 0.304 |
| Intercept | $\hat{\mu}_t^{(k_0)}(s) \in \hat{\boldsymbol{\beta}}_t^{(k_0)}(s) \subset \hat{\boldsymbol{\theta}}^{(k_0)}$ | 14.067 ** (0.464) | 6.273 ** (0.050) | 0.673 ** (0.080) |
| Economic activity index coefficient | $\gamma_{S_t}^{k_0} \in \hat{\boldsymbol{\beta}}_t^{(k_0)}(s) \subset \hat{\boldsymbol{\theta}}^{(k_0)}$ | 0.0124 (0.302) | 0.236 ** (0.052) | 0.075 ** (0.032) |
| Interest rate coefficient | $\delta_t^{(k_0)}(s) \in \hat{\boldsymbol{\beta}}_t^{(k_0)}(s) \subset \hat{\boldsymbol{\theta}}^{(k_0)}$ | −1.381 ** (0.388) | −0.695 ** (0.067) | −0.239 ** (0.042) |
| Iterations for convergence Iterations considered | $k_0 = 92$ $k_{max} = 500$ | | | |

** Significative at 95% of confidence level.

The expansive state converged to an annual real property value index growth rate of 6.3%, practically twice the estimated rate for the average growth regime. Likewise, it can be observed that when real property value index grew expansively, it exhibited greater volatility or dispersion than other alternative states of nature. This greater dispersion generated overlaps in some segments of the distributions of the different states, hindering the precision of their identification in the absence of statistical tools (Figure 5). In relation to the variables that condition this real property value index behavior, the importance of the mortgage interest rate stood out with respect to the economic activity index impetus. In other words, the results of the switching model show that episodes of high housing prices were mainly explained by favorable financial conditions for long-term loans with respect

to short-term economic performance (measured by economic activity index). This finding emerged from comparing the magnitude and significance of the coefficients that determine the expansive state of the annual real property value index growth rate.

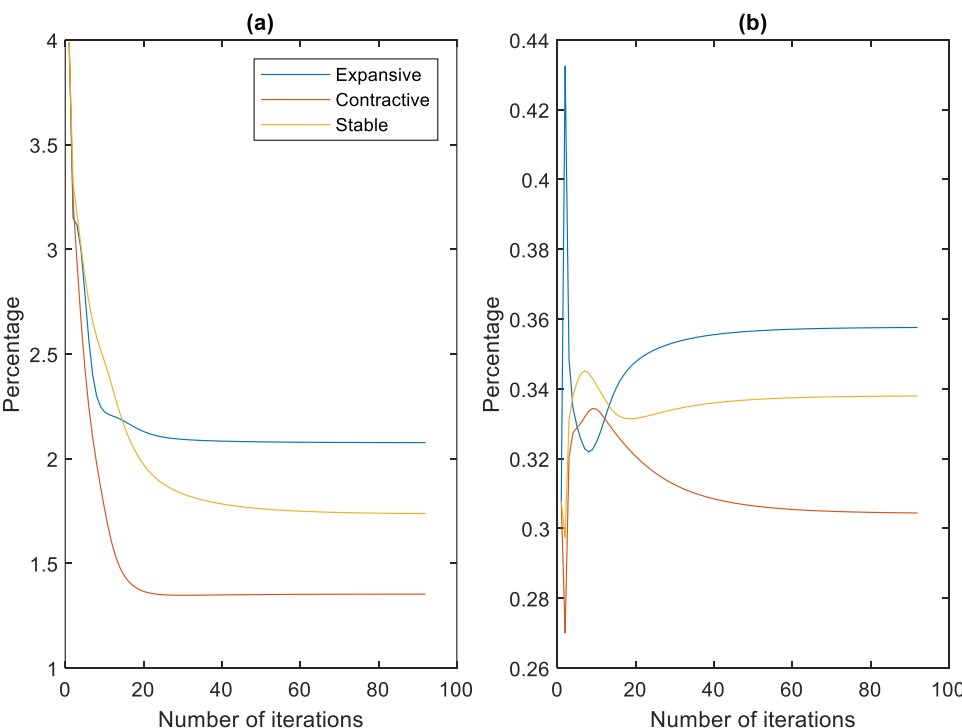

**Figure 5.** Evolution of the convergence rates of the EM algorithm. (**a**) Standard deviation (volatility) of annual real property value index growth (in %); (**b**) probability of observing the alternative state through which the real property value index is passing.

The contractionary state of the real housing price converged to the negative rate of 0.4% in annual variation, with a volatility of 1.4%. Under this regime, the price dynamics were consistent with fundamental macroeconomic variables, since economic activity index and interest rate were significant at 95% of confidence level. Therefore, the results of this switching model confirm that the relationship between the housing price and its fundamentals was asymmetric and conditional on the real estate sector's original cycle, which, as shown above, was a characteristic found in the literature on housing prices and property booms.

In summary, the multipolarity observed in the annual growth of real property values was partly explained by its non-linear response to economic activity and the behavior of the long-term interest rate. Therefore, in hindsight, the real property value index evolution has been consistent with fundamental macroeconomic variables, even under a high growth regime—with price increases that have exceeded 12% per year. Another point to highlight is that the relevance of financial conditions for access to mortgage credit observed within each alternative state constituted a common element to explain the heterogeneity in the trajectory of the housing price in Chile.

Figure 6 shows the convergence evolution of the standard deviation (or volatility) of the real property value index ($\hat{\sigma}_{s_t}^2$) for each alternative state ($S_t$). It also shows the convergence path of the probability measures as the housing price passed through one of these growth regimes $\mathrm{E}[\mathbb{P}(S_t|\hat{\boldsymbol{\theta}}^{(k_0)})]$. The probability of observing an expansive state of housing price growth on average is 0.36, somewhat higher than that estimated for the other two alternative states (medium and contractive). Therefore, the real property value index growth rate has a certain upward bias.

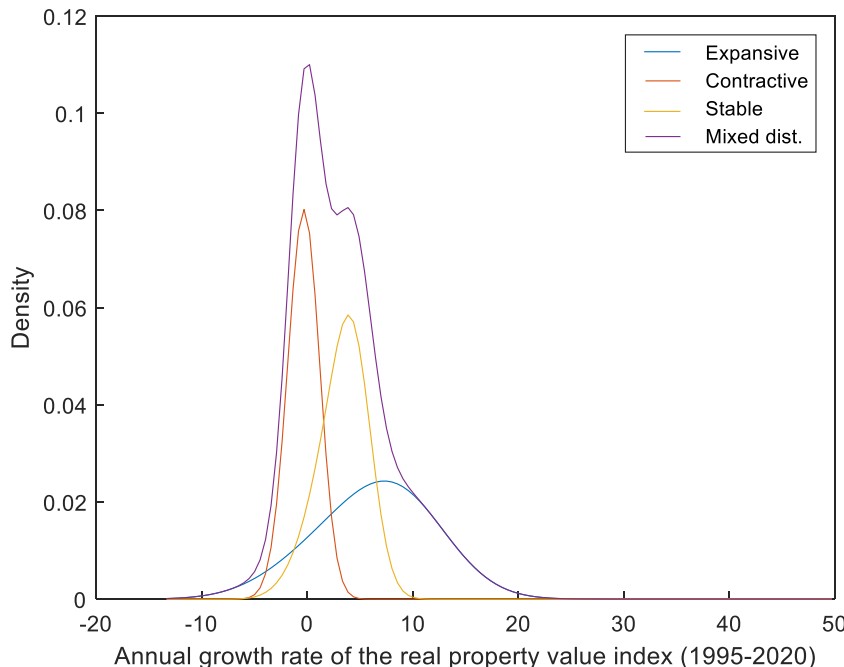

**Figure 6.** Mixed distribution of the annual growth rate of the real property value index (1994–2020). Distribution function of real property value index annual growth and its alternative states.

The foregoing is consistent with the high housing price growth from 2011 to 2015 and 2018 to 2019, when some analysts warned of a real estate bubble in Chile (Gil-Alana et al. 2019; Smart and Burgos 2018). However, Idrovo and Lennon (2012, 2013), BCCh (2018), and Silva and Vio (2015) showed that this price growth was mostly due to favorable macrofinancial investment conditions in this market. Added to this scenario was the anticipated effect on the demand after the government's announcement to impose VAT on home sales. These elements allowed us to statistically rule out the relevance of speculative behaviors in the dynamics of real estate prices up to 2018. Although, these findings were based on statistical tests contrasting the existence of long-term linear relationships between housing prices and their fundamental variables. The results of our regime change model corroborate a synchronization or alignment of real property value index with its macroeconomic determinants, annual economic activity index growth, and mortgage interest rate, in the short and long term.

One advantage of estimating a regime change model is that it allows adjusting in a simple way non-linear short-term relationships implicit in the growth rate of the housing price through the identification of alternative regimes or states of their behavior. Thus, considering the estimated parameters for the three growth regimes (Table 4), it is possible to graph the distribution mixture of the housing price growth rate conditional on its fundamental variables. This result is visualized in Figure 5, showing that the distributions of the alternative states of the real property value index growth rate overlap in some segments. Therefore, the simple observation of the fluctuations of the housing price is not enough to achieve the precise identification of these regimes, which can lead to erroneous conclusions regarding the real situation of the real estate market.

Table 5 shows the transition probabilities between the different regimes, i.e., the probability that the annual real property value index growth rate will remain in the same state or migrate to an alternative state in the short term. In addition, these probabilities provided information on the length of home price cycles (Kim and Nelson 1999). For example, the average duration of growth cycles in housing prices was 6.1 months, and the probability that the price remained in this regime for two consecutive months was 0.84. On the other hand, the probabilities that the housing price migrated from a state of medium growth to an expansive or contractionary cycle were 0.13 and 0.11, respectively. Furthermore, historically these transitions were almost instantaneous, since on average they have a latency period of 1.1 months. Again, these results coincide with the evidence

in the literature about the asymmetry of real estate cycles in terms of their duration and depth, as discussed above.

**Table 5.** Transition probabilities and average duration of the alternate stages of real property price growth. The values in brackets correspond to the average duration of the real property value index cycles, considering their historical evolution.

|  | Expansive $(S_{t-1} = 1)$ | Medium $(S_{t-1} = 2)$ | Contractive $(S_{t-1} = 3)$ |
|---|---|---|---|
| Expansive $(S_t = 1)$ | $p_{11} = 0.836$ [6.1 months] | $p_{12} = 0.135$ [1.2 months] | $p_{13} = 0.043$ [1.0 months] |
| Medium $(S_t = 2)$ | $p_{21} = 0.127$ [1.1 months] | $p_{22} = 0.768$ [4.3 months] | $p_{23} = 0.108$ [1.1 months] |
| Contractive $(S_t = 3)$ | $p_{31} = 0.037$ [1.0 months] | $p_{32} = 0.097$ [1.1 months] | $p_{33} = 0.849$ [6.6 months] |

*Real Property Value Index Simulation and Probability Index*

The real property value index simulation and the construction of the housing price exuberance probability index were based on conditional probabilities $\mathbb{P}(S_t | \tilde{\mathbf{y}}_T, \hat{\boldsymbol{\theta}}^{(k)})$, obtained from the estimation of the regime change model. Figure 7 shows the historical evolution of the probabilities conditional on the three predefined states of the annual real property value index growth: expansive, medium, and contractive. The average values of these observations corresponded to the long-term probabilities for each alternative state shown in Table 4.

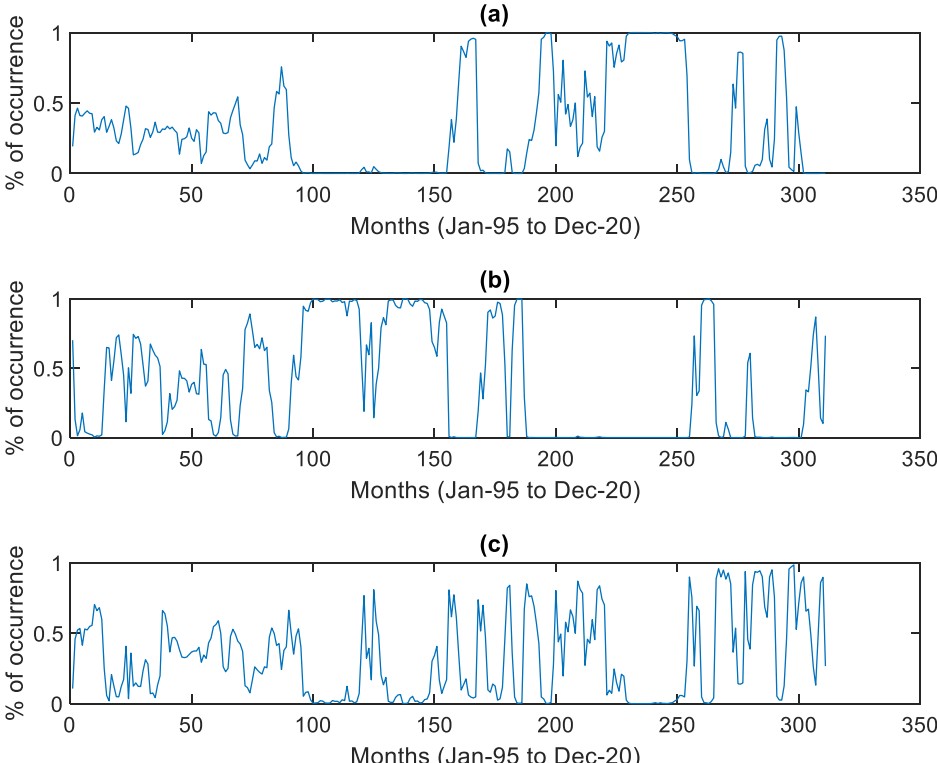

**Figure 7.** Conditional probabilities for each state of annual real property value index growth (January 1995 to December 2020). (**a**) Conditional probabilities for the expansive regime, (**b**) the medium growth regime, (**c**) the contractive regime.

The simulation of the housing price was obtained from the weighted sum of the alternative states with their predetermined macroeconomic variables, where the weights

corresponded to each state's conditional probabilities. In other words, the simulated price is the expected real property value index value conditional on its three annual growth regimes and its fundamental variables. Likewise, the price variance corresponds to a weighted average of its growth volatility in each state

$$\hat{p}_t = \sum_{j=1}^{3} \mathbb{P}(S_t = j | \tilde{\mathbf{p}}_T, \hat{\boldsymbol{\theta}}^{(k)})(\mathbf{x}_{t-1}\hat{\boldsymbol{\beta}}^{(k)}_{s_t=j}), \tag{20}$$

$$\hat{\sigma}_p^2 = \sum_{j=1}^{3} \mathbb{P}(S_t = j | \tilde{\mathbf{p}}_T, \hat{\boldsymbol{\theta}}^{(k)})(\hat{\sigma}_t^{2(k)}(s)), \tag{21}$$

where $\hat{p}_t$ is the simulated housing price, $\mathbb{P}(S_t | \tilde{\mathbf{p}}_T, \hat{\boldsymbol{\theta}}^{(k)})$ is the conditional probability of each state $j$ with $k$ optimal iterations that assures the expectation–maximization algorithm application, $\hat{\boldsymbol{\beta}}^{(k)}_{s_t=j}$ is the parameter vector relating the actual housing price with its fundamental variables in each alternative state $j$. $\mathbf{x}_{t-1}$ is the vector of predetermined variables of the regime change model, the $\hat{\sigma}_p^2$ parameter is the conditional variance of the housing price growth rate, and $\hat{\sigma}_t^{2(k)}(s)$ measures the scattering of the price growth within each state $j$. From definition $\hat{p}_t$ and $\hat{\sigma}_p^2$ a confidence interval for the simulation of the real property value index was constructed. Figure 8 compares the real property value index trajectory with its simulated values based on the estimated parameters of the regime change model. In Figure 8a the resulting determination coefficient was $R^2 = 0.915$. This means that 92% of the annual growth variance of real property value index was explained by the model. Figure 8b shows that the observed real property value index growth rate fluctuated within the 95% confidence band estimated by the model.

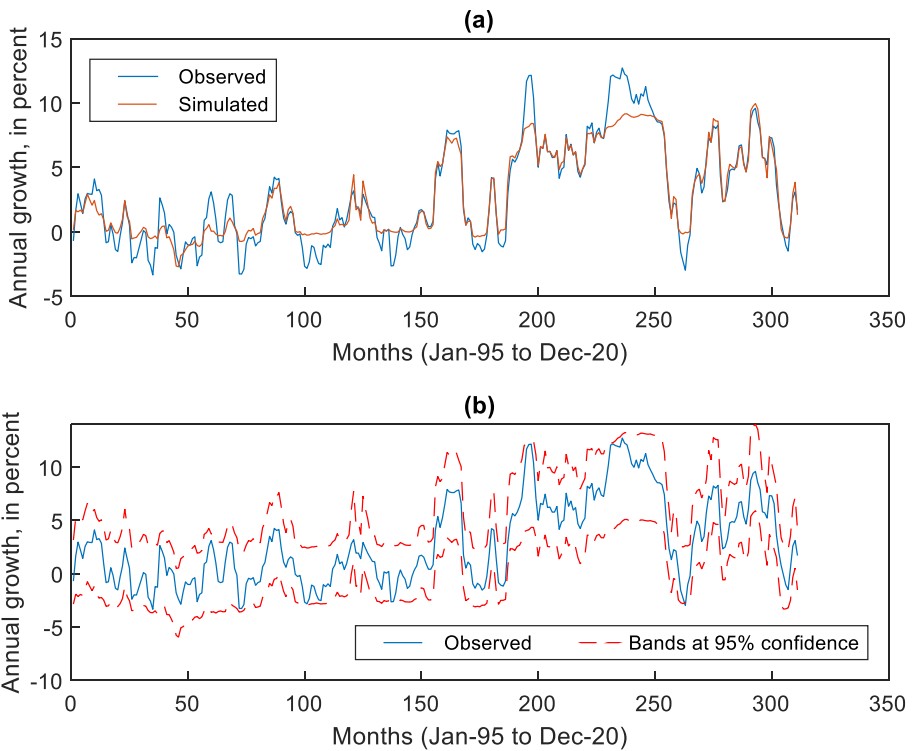

**Figure 8.** Real property value index evolution and its simulation based on the estimated parameters of the regime change model. (**a**) Observed and simulated real property value index; (**b**) observed real property value index and a 95% confidence interval of its simulation.

This can also be seen in Figure 9, which shows the evolution of the real housing price level with its simulated version based on the switching model parameters with three alternative states of nature and two predetermined macroeconomic variables (economic

activity index and real mortgage interest rate). This way, the stochastic trend process that the real property value index appeared to follow was replicated through a model with mean and variance corresponding to each regime. Table 6 shows that the real housing price expressed in level and annual variation contained a unit root at 1% and 5% of significance, while the series of the estimated residual of the switching model followed a weakly stationary process, according to augmented Dickey–Fuller (Dickey and Fuller 1979) and Phillips–Perron (Phillip and Perron 1988) tests.

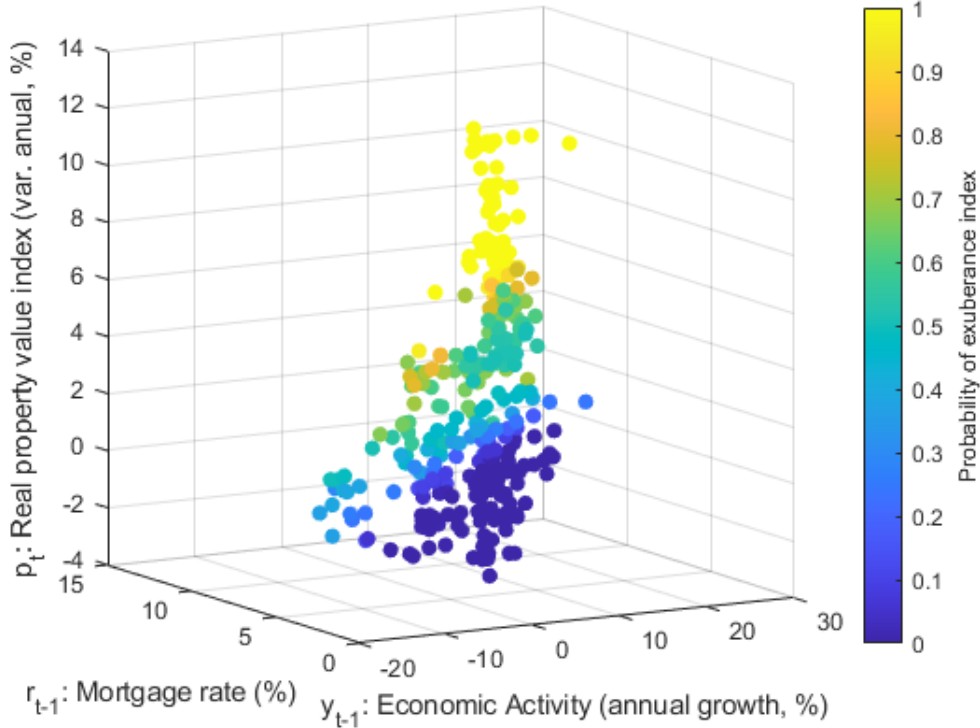

**Figure 9.** Real property value index: Observed vs. simulated.

**Table 6.** Unit root statistics tests. (a) Equation includes only constant, (b) equation includes constant and trend. DFA: Augmented Dickey–Fuller test. PP: Phillips–Perron test. $P_t$ is the real property value index at time $t$; $p_t$ is the annual real property value index growth rate at time $t$; and $y_{t-1}$ is the annual economic activity index growth rate in $t-1$. $r_{t-1}$ is the mortgage interest rate in $t-1$ and $\hat{\varepsilon}_t$ is the estimated series of the switching model residuals.

| Indicators | Levels (a) | | Levels (b) | |
|---|---|---|---|---|
| | DFA | PP | DFA | PP |
| Real property value index $(P_t)$ | 3.30 | 3.62 | −0.25 | 0.14 |
| $\Delta^{12}$ Real property value index $(p_t)$ | −1.86 | −3.07 | −2.54 | −3.69 |
| Economic activity index $(y_{t-1})$ | −4.34 | −4.16 | −5.05 | −4.89 |
| Interest rate $(r_{t-1})$ | −1.21 | −0.89 | −2.35 | −2.01 |
| Residuals $(\hat{\varepsilon}_t)$ | −5.29 | −5.93 | −5.34 | −5.92 |
| Critical Value | | | | |
| 1% | −3.45 | −3.45 | −3.99 | −3.99 |
| 5% | −2.87 | −2.87 | −3.42 | −3.42 |

Based on the data, housing prices in Chile have evolved according to key macroeconomic determinants. Therefore, a priori any existence of a real estate bubble in the historical evolution of real property value index is ruled out, in line with the findings of Idrovo and Lennon (2012, 2013) and Silva and Vio (2015), who corroborated the existence of cointegration between the real estate price and some market supply and demand factors.

Along these lines, the BCCh (2018) concluded in its Financial Stability Report that there was insufficient statistical evidence to support the presence of some fundamental misalignment in the price-income ratio from 2006 to 2016. This is based on the use of statistical tests to contrast differences between the evolution of housing prices and household income, following the methodology of Martinez-García and Grossman (2020).

The exuberance probability index of the real housing price was built based on the methodology used by Johnson (2001) for the creation of a monetary policy modification index. However, the variant here is that this methodology is applied to the annual growth of real property value index and included the predetermined switching model or explanatory variables of housing prices. From the previously analyzed conditional probabilities, the real property value index exuberance index was built, whose range of values fluctuates between 0 and 1. This indicator is conditioned by the economic activity cycle—proximate to annual economic activity index growth—and the access conditions to financing, collected in the annual interest rate of mortgage loans. A value close to 1 signals the housing price going through a cycle of exuberance, due to an overheating of economic activity and/or favorable access conditions for long-term credit. As the probability index tends to 1, risks to financial stability increase, suggesting the necessity for economic policy adjustments. If the index registers values close to 0.5, the real estate market does not present risks to financial stability. Finally, if the indicator approaches zero, the risk of contraction emerges. This could be partially explained by a weakening of economic activity or restrictive financial conditions in the real estate market. So according to the model, the contractionary scenario could stem from weakening demand for durable goods such as housing and/or a worsening of credit flow. This, among other factors, could suggest the need to adopt expansionary economic policy.

In formal terms, Johnson (2001) proposed a sigmoidal transformation of the conditional probability ratio to reach an alternative state for the construction of the probability index. The following expression summarizes the objective indicator:

$$\Psi_t = \frac{e^{\nu_t}}{1 + e^{\nu_t}}, \tag{22}$$

$$\nu_t = \frac{\mathbb{P}(S_t = 1|\tilde{\mathbf{y}}_T, \hat{\boldsymbol{\theta}}^{(k)}) - \mathbb{P}(S_t = 3|\tilde{\mathbf{y}}_T, \hat{\boldsymbol{\theta}}^{(k)})}{\mathbb{P}(S_t = 2|\tilde{\mathbf{y}}_T, \hat{\boldsymbol{\theta}}^{(k)})}, \tag{23}$$

where $\Psi_t$ it is the real property value index exuberance probability index, which takes real values in the range from 0 to 1; the argument $S_t = 1$ refers to the expansionary regime of the annual growth rate of the real housing price, and $\mathbb{P}(S_t = 1|\tilde{\mathbf{y}}_T, \hat{\boldsymbol{\theta}}^{(k)})$ is the probability that the price goes through the regime. The argument $S_t = 2$ corresponds to the average state of price growth, and $\mathbb{P}(S_t = 2|\tilde{\mathbf{y}}_T, \hat{\boldsymbol{\theta}}^{(k)})$ is the probability function of occurrence. Finally, the argument $S_t = 3$ corresponds to the contractive state of annual growth of real property value, with a probability of occurrence of $\mathbb{P}(S_t = 3|\tilde{\mathbf{y}}_T, \hat{\boldsymbol{\theta}}^{(k)})$. Figure 10 shows the trajectory of the probability index proposed here and the annual real property value index growth rate.

Figure 10 shows that the results of the probability index are consistent with real property value index fluctuations from January 1995 to December 2020. Particularly, as of June 2008, some states of exuberance in housing prices emerged. The June 2013–December 2015 period stands out due to the persistence of the index in registering values around 1. This regime was not only marked by greater economic activity and low interest rates but also by the positive impact the advance announcement of incorporating VAT had until January 2016. Figure 11 shows that these levels of exuberance in Chile were consistent with the economic and financial conditions of the period, discarding a priori preponderant participation of speculative behaviors in the historical evolution of real property value index.

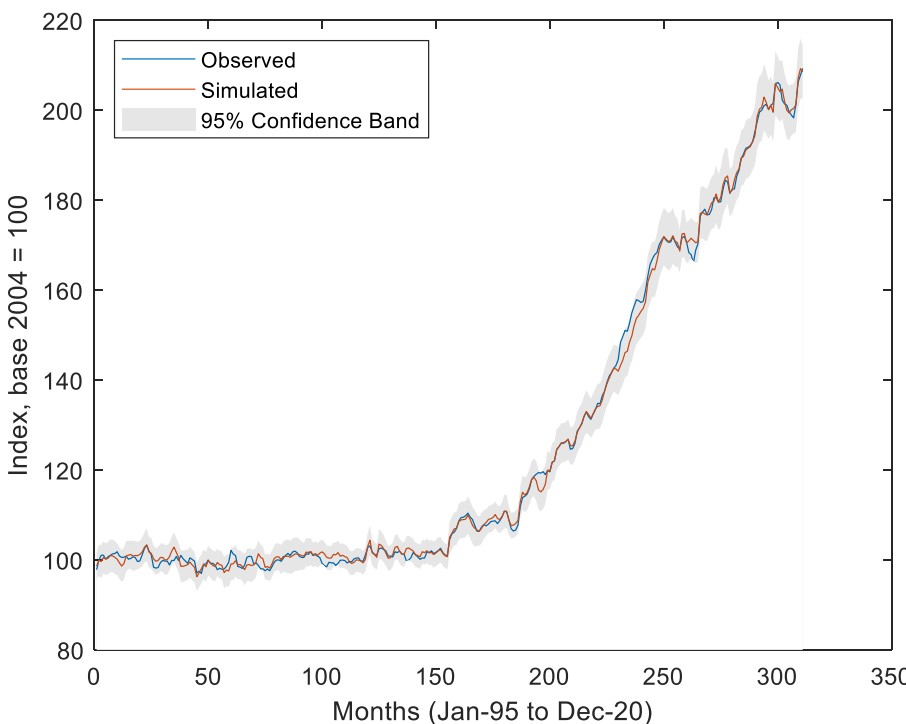

**Figure 10.** Evolution of the annual real property value index growth rate and its exuberance probability index.

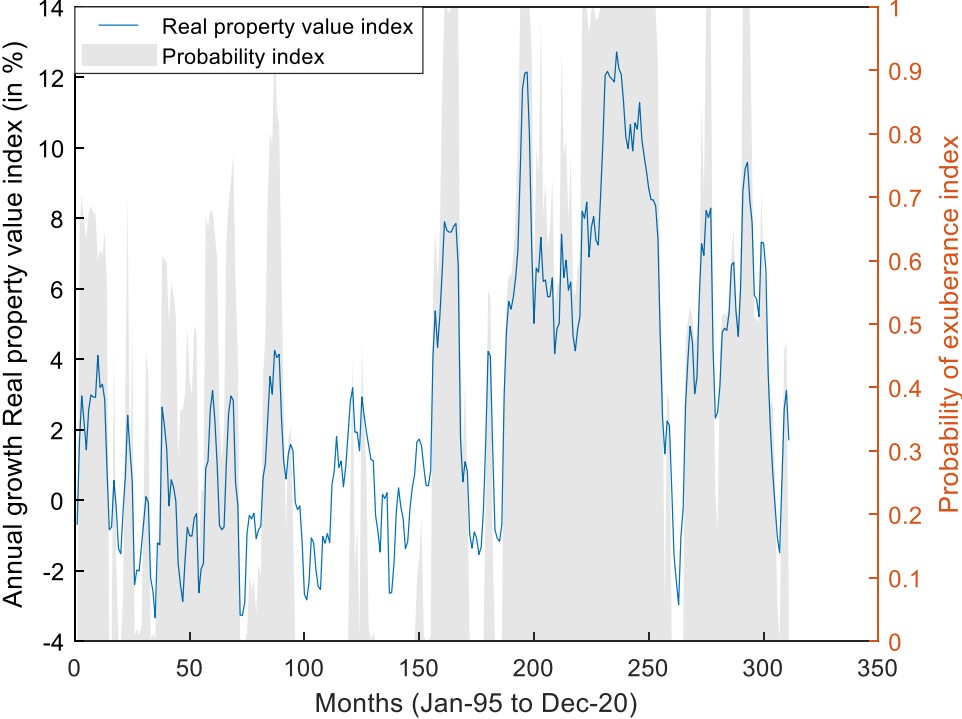

**Figure 11.** New real property value index exuberance index and its macroeconomic determinants.

In summary, the results obtained when estimating a Markov regime switching model of changing regimes highlighted the importance of economic and financial conditions to explain housing price evolution. These two factors have been common in the international literature (Adams and Füss 2010; Anundsen and Jansen 2013; Chen et al. 2014; Elbourne 2008; Goodhart and Hofmann 2008; Helbling 2005). In the case of Chile, we have found

a relevant adjustment between these explanatory factors and increasing housing prices, which leaves little room for speculative behavior that might be the origin of a real estate bubble in the local market (Aguilera Alvial 2020; Gil-Alana et al. 2019; Idrovo and Lennon 2012 2013; Silva and Vio 2015). The most appropriate model in this case would consider expansive, medium, and contractive states, consistent with other results in the international literature (Nneji et al. 2013; Prüser and Schmidt 2021; Rangel and Ng 2017; Sethapramote et al. 2019). Finally, in line with other stylized facts in the literature on non-linear relationships in the real estate sector, asymmetries were found among the states that govern the housing price movements. First, asymmetry with respect to volatility in each state, the expansive state exhibiting the greatest dispersion in annual growth rates (Cerón and Suárez 2006; Chen et al. 2014; Chowdhury and Maclennan 2015; Espinosa and Sanin 2016; Kim and Min 2011); and second, asymmetry regarding the marginal effect of each explanatory factor as economic activity has greater impact in medium and contractive states, financial conditions have a greater impact in expansionary and medium states (Chowdhury and Maclennan 2015; Espinosa and Sanin 2016; Nneji et al. 2013; Rangel and Ng 2017; Savva 2015).

## 5. Conclusions

The economic literature on Chile's housing market focused on estimating linear relationships between the prices and their economic and financial determinants to find possible deviations from a long-term equilibrium. Although these models have served to understand the recent evolution of the real estate sector, a significant knowledge gap remains regarding new estimation methods. As a novelty, this work applied a model of non-linear relationships between the housing price and its economic foundations. The methodology is also widely found in the international literature.

We identified a non-linear relationship between the real property value index, the economic activity index, and the interest rate for mortgage loans for the 1994–2020 period. We developed a regime change model with three states of nature and with exogenous or predetermined variables (second-order Markov switching model with predetermined variables), whose parameters were estimated through the iterative expectation–maximization algorithm. The states for the price growth rate were expansive, medium, and contractive. The average growth rate (or long-term growth) of real property value index was 3.3% per year, similar to the inflation target of 3% per year established by the Central Bank. The expansive state converged to a 6.3% annual growth rate of real property value index and its volatility or dispersion (2.1%) exceeded that of other alternative states of nature, 1.7% and 1.4% for the medium and contractive states, respectively. This greater dispersion generated overlaps in some segments of the distributions of the different states, hindering the precision of their identification in the absence of statistical tools. Episodes of high housing prices were mainly explained by favorable financial conditions for long-term loans with respect to short-term economic performance. This is consistent with international evidence about the relationship between credit expansion and real estate market booms. Meanwhile, the contractionary state of the real housing price converged to the negative rate of 0.4% of annual variation.

One of our main findings was that ex post high housing price events in Chile, especially from 2008 onwards, have been consistent with the macrofinancial situation of the period. Therefore, a priori the existence of any predominant speculative behaviors could be ruled out. For example, the pace of exuberant real housing price growth between 2013 and 2015 rested on a combination of favorable higher economic activity and low mortgage interest rates in inflation-linked units. This scenario, in part, contributed to higher real estate demand from households and investors, particularly after the government announced intentions of imposing value added tax (VAT) on home sales.

Although each regime has an associated probability of occurrence, the probability of observing an expansive state of price growth in the long term is 0.36, somewhat higher than figures estimated for the other two alternative states. Therefore, the real property value index growth rate had a certain upward bias. With these measures we set up a probability index to function as an early warning indicator of potential imbalances in the real estate price that could put financial market stability at risk. The indicator would be relevant to timely evaluating possible economic policy calibrations. The probability index

shows that the evolution of the real estate price has been consistent with its fundamental macroeconomic variables, even under a high growth regime with increases that exceed 12% per year. Therefore, the episodes of high housing prices in Chile, especially from 2008, have been consistent with the macrofinancial situation of the period. For example, the exuberant rate of growth in real housing prices that we observe between 2013 and 2015 was due to a favorable combination of greater economic activity and low mortgage interest rates. This scenario, in part, contributed to the higher demand for real estate from households and investors, particularly after the government announced it will impose VAT on home sales.

Building a land price index constitutes a future task, since no such indicator currently exists. This could be relevant to quantitatively identify the response of the real housing price to land value, as indicated by the Central Bank in its 2018 Financial Stability Report (BCCh 2018). On the other hand, the need arises to construct qualitative or quantitative variables that allow directly measuring the effect of laws or regulations have on land use in the real estate market. A priori, the construction of these regulation variables could be based on the methods or algorithms for searching and selecting keywords in the government's official gazette, where all legal norms are being published. The methodology proposed by Baker et al. (2016) and used by Cerda et al. (2016) for the creation of an uncertainty index could be appropriate for this purpose. This set of new variables, not yet available for Chile, could be added to the current determinants of economic activity index and the mortgage interest rate used here.

**Author Contributions:** B.J.I.-A., F.J.L. and J.E.C.-R. wrote the paper and contributed reagent/ analysis/material tools; B.J.I.-A. conceived, designed, and performed the experiments and analyzed the data. All authors have read and agreed to the published version of the manuscript.

**Funding:** This research was funded by FONDECYT (Chile) grant No. 11190116.

**Institutional Review Board Statement:** Not applicable.

**Informed Consent Statement:** Not applicable.

**Data Availability Statement:** Dataset available at http://dx.doi.org/10.13140/RG.2.2.21199.61601 (accessed on 28 May 2021).

**Acknowledgments:** Contreras-Reyes's research was fully supported by FONDECYT (Chile) grant No. 11190116. The authors thank the editor and two anonymous referees for their helpful comments and suggestions. All R codes used in this paper are available upon request from the corresponding author.

**Conflicts of Interest:** The authors declare no conflict of interest.

## Note

1   The advantage of these models over structural change models proposed by Chow (1960) is that the times in which the regime changes occur are endogenous to the model.

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
