# Peer review of "Prosperity or Real Estate Bubble? Exuberance Probability Index of Real Housing Prices in Chile"

_ijfs, doi:10.3390/ijfs9030051_

Round 1
Reviewer 1 Report
Typeset by LATEX using class file mdpi.cls in page 1 is not needed
Please copy edit the whole document again by a native English speaker, e.g. Introducción should be Introduction.
Abstract
We analyzed the relationship between real housing prices, the Imacec economic activity index, and mortgage loan interest rates denominated in UF (local inflation-linked unit)...
This one is quite old concept, please state originality, academic and practical contribution.
A main finding was that the real housing price had a non-linear relationship
with economic activity and the mortgage interest rate. This finding is not interesting at all (many similar research available now).
p.2, first paragraph, proposed by Hamilton (1990), check if this is an appropriate method to cite.
please reduce the number of abbreviations such as RPVI and UF, we cannot expect our readers check and reread over again all the time.
p.5, One is the mortgage interest rate in UF for housing (or real
mortgage interest rate), which contains information on the financial conditions for access to housing. The other is Imacec activity. The Imacec index measures the average income behavior of agents’ that
participate in this market.
It seems to me that these are very common types of factors in housing price models.
Please add a section of data description and summary before the results and after the method.
Please cite the paper https://www.mdpi.com/2071-1050/10/2/341 and consider if that is possible to add some more factors of higher originality that lead to greater contribution to scholarship.
Replace some of the non-journal articles by journal articles.
Author Response
- Typeset by LATEX using class file mdpi.cls in page 1 is not needed.
R: Done as suggested.
- Please copy edit the whole document again by a native English speaker, e.g. Introducci´on should be Introduction.
R: The manuscript have been send to a native English editor, including the new inserted paragraphs.
- Abstract: We analyzed the relationship between real housing prices, the Imacec economic activity index, and mortgage loan interest rates denominated in UF (local inflation-linked unit)... This one is quite old concept, please state originality, academic and practical contribution. A main finding was that the real housing price had a non-linear relationship with economic activity and the mortgage interest rate. This finding is not interesting at all (many similar research available now).
R: Abstract was modified in the first lines according your observation and to give the novelty of the paper.
- p.2, first paragraph, proposed by Hamilton (1990), check if this is an appropriate method to cite.
R: We agree with your observation, thus ”proposed by Hamilton (1990)” was deleted and Ref. [17] is kept.
- Please reduce the number of abbreviations such as RPVI and UF, we cannot expect our readers check and reread over again all the time.
R: RPVI, UF, EM, Imacec and PVI appears in completely form as suggested.
- 5, One is the mortgage interest rate in UF for housing (or real mortgage interest rate), which contains information on the financial conditions for access to housing. The other is Imacec activity. The Imacec index measures the average income behavior of agents’ that participate in this market. It seems to me that these are very common types of factors in housing price models.
R: In order to give more emphasis of the paper’s novelty, we modified the abstract and inserted the lines 508-522 of Conclusion section according your observation.
- Please add a section of data description and summary before the results and after the method. R: Section 3 was moved as suggested.
- Please cite the paper https://www.mdpi.com/2071-1050/10/2/341 and consider if that is possible to add some more factors of higher originality that lead to greater contribution to scholarship.
R: Suggested reference added as suggested. In addition, suggested factors have been included as further work at lines 523-532.
- Replace some of the non-journal articles by journal articles.
R: Several new references (marked in red in References section) have been included. About non-journal articles: in Chile, the actual economic literature have used linear models for factor identification that conditioned the housing prices, such as economic activity index and mortgage interest rate. In particular, VEC models are the most used by researchers to account the relationship of price cointegration with their fundamental variables. However, the most of these works are in Working Paper format (non-journal). But we considered now papers such as Aguilera (2020) and others marked in red.
With kind regards.
Reviewer 2 Report
Thank you for providing me with the opportunity to read “Prosperity or real estate bubble? Exuberance probability index of real housing prices in Chile”. I have the following comments to improve the paper:
- Please provide the names (or some of them, if a lot) for the predetermined variables in the abstract.
- Please spell out EM and all other abbreviations on the first appearance
- Markov switching is a keyword and not mentioned at all in the abstract. Is it not important?
- The formation of real estate bubbles may not always have negative consequences. It is important to clarify that the authors are focusing on the downside (threats) of risks.
- The authors need to establish the context for this study in the introduction before mentioning the purpose of the study. The authors need to clarify what exactly the problem is and why this study is needed. Afterward, the purpose can be presented
- The novelty of the study must be highlighted in a detailed paragraph and compared with existing studies to show what innovation is brought in by this study. This can be added after the last paragraph on page 1.
- The authors need to add more and recent references to the paper. Most of the references are outdated, please use more references from recent years this will also help you get more familiarity with the recent development in this area
- Please add references to the first para of the literature section. The same comment for the second last paragraph. Provide references for the lists of factors/financial conditions and economic activity.
- The Exuberance probability must be discussed in detail, and studies utilizing it highlighted and discussed.
- Please provide the weblink to the “Personal income tax figures” published by the Chilean Chamber of Construction. Same for the data where the Central Bank of Chile data can be viewed.
- Any reference claiming the four well-marked periods of Chile’s economic cycle? Please provide the reference to justify this statement
- The predetermined variables need to be discussed in detail and listed along with references in a separate table to show how and why these are important
- Have any previous studies used the Em algorithm? Please compare the innovation of this algorithm in the current study with previous studies and discuss how this model is innovative or different in the current study
- Please improve the discussion of the study and add a section at the end of the result to compare the key findings of this study with existing works and highlighting the key contributions to the body of knowledge and practice.
- Please add the precise limitations of the study to the conclusion section.
Thanks and all the best
Author Response
Thank you for providing me with the opportunity to read ”Prosperity or real estate bubble? Exuberance probability index of real housing prices in Chile”. I have the following comments to improve the paper:
- Please provide the names (or some of them, if a lot) for the predetermined variables in the abstract.
R: Sentence ”Personal income tax figures” fixed according your observation.
- Please spell out EM and all other abbreviations on the first appearance.
R: Done. Also EM was spell out in all manuscript as suggested by both referees.
- Markov switching is a keyword and not mentioned at all in the abstract. Is it not important?
R: Done. ”Markov switching of 2nd order with predetermined variables” was inserted in abstract.
- The formation of real estate bubbles may not always have negative consequences. It is important to clarify that the authors are focusing on the downside (threats) of risks.
R: Please consider the new lines 11-20 (in Abstract) and 131-136 (in Introduction section).
- The authors need to establish the context for this study in the introduction before mentioning the purpose of the study. The authors need to clarify what exactly the problem is and why this study is needed. Afterward, the purpose can be presented.
R: Introduction restructured as suggested.
- The novelty of the study must be highlighted in a detailed paragraph and compared with existing studies to show what innovation is brought in by this study. This can be added after the last paragraph on page 1.
R: Please consider the new lines 11-20 (in Abstract) and 508-522 (in Conclusions).
- The authors need to add more and recent references to the paper. Most of the references are outdated, please use more references from recent years this will also help you get more familiarity with the recent development in this area.
R: We have included new (and recent) references (see references marked in red) following your suggestion, e.g. Aguilera (2020).
- Please add references to the first para of the literature section. The same comment for the second last paragraph. Provide references for the lists of factors/financial conditions and economic activity.
R: Following your suggestion, we have included new references such as: Mart´Ä±nez and Oda (2021), Aguilera (2020), and Idrovo-Aguirre and Contreras-Reyes (2019, 2021).
- The Exuberance probability must be discussed in detail, and studies utilizing it highlighted and discussed.
R: Exuberance of housing price is a term used in Central Bank financial stability report, based on Mart´Ä±nez et al. (2018) (which appears cited in our paper). This term is related to untenable housing price growth related to their fundamental variables, such as the economic cycle and mortgage interest rate. Therefore, Exuberance probability index accounts to probability of fundamental misalignment of housing price that could give risk the stability of financial market of Chile. This has been included in Introduction and Methodology sections.
- Please provide the weblink to the ”Personal income tax figures” published by the Chilean Chamber of Construction. Same for the data where the Central Bank of Chile data can be viewed.
R: Concept ”Personal income tax figures” was fixed. Also, we included the webpage of several references (Working Papers and data sources).
- Any reference claiming the four well-marked periods of Chile’s economic cycle? Please provide the reference to justify this statement.
R: References added as suggested at lines 233-235.
- The predetermined variables need to be discussed in detail and listed along with references in a separate table to show how and why these are important.
R: Table 2 has been included according your suggestion.
- Have any previous studies used the Em algorithm? Please compare the innovation of this algorithm in the current study with previous studies and discuss how this model is innovative or different in the current study. Please improve the discussion of the study and add a section at the end of the result to compare the key findings of this study with existing works and highlighting the key contributions to the body of knowledge and practice.
R: New lines 508-522 have been included in Conclusions for comparisons as suggested.
14. Please add the precise limitations of the study to the conclusion section.
R: New lines 523-532 have been included in Conclusions as suggested.
With kind regards
Round 2
Reviewer 1 Report
The revised version is a lot better than the previous version, I would like to suggest it to get published after some minor revisions, in particular, the results sections, where some headings to highlight the main findings are necessary to ease readers in figuring out what the authors wish to convey, and what are the main contributions out from these passages. Other comments are as follows:
First paragraph, missing citation.
Line 57, remove unnecessary space before But.
Lines 284-285, we considered two fundamental macroeconomic variables of the real estate market to explain real property value index dynamics. Please use literatures to support.
Table 2, please turn the sentence to English.
conclusions, state the gap that it tries to fill.
What does that tell for non linear relationships.
a monthly real land price index may not be useful, unless there are land sales all the times.
References 11-13 and other non-English references, please use English with bracket at the end (in Spanish), for example.
Copy edit the paper. Minor revision is needed.
Author Response
1. First paragraph, missing citation.
R: References added as suggested.
2. Line 57, remove unnecessary space before But.
R: We change "But" by "However".
3. Lines 284-285, we considered two fundamental macroeconomic variables of the real estate
market to explain real property value index dynamics. Please use literatures to support.
R: Reference Johnson (2001) was inserted as suggested.
4. Table 2, please turn the sentence to English.
R: Apologize for this mistake. Sentence was turned to English as suggested.
5. Conclusions, state the gap that it tries to ll.
R: New lines inserted at lines 502-507 according your suggestion.
6. What does that tell for non linear relationships.
R: New lines inserted at lines 505-507 according your suggestion.
7. A monthly real land price index may not be useful, unless there are land sales all the times.
R: E ectively, it could be complicated to have a land price index with monthly frequency
given the low volume of transactions. However, there exist methodologies to solve this task,
for example, mobile samples such as we used the real property value index. Therefore, in
line 553, we removed the word "monthly" to have "land price index".
8. References 11-13 and other non-English references, please use English with bracket at the end
(in Spanish), for example.
R: Spanish literature have been translated to English as suggested.
With kind regards.
Reviewer 2 Report
The authors have addressed most of my comments however a response and pertinent addition to the paper are required for the following before the paper can be accepted.
- The authors need to clarify what exactly the problem is and why this study is needed. This is yet to be seen in the paper
- The novelty of the study must be highlighted in a detailed paragraph and compared with existing studies to show what innovation is brought in by this study. This can be added after the last paragraph on page 1. Please note this is required in the introduction to justify why this study is needed.
- Please add a discussion section at the end of the result to compare the key findings of this study with existing works. The authors have provided some addition in conclusions but as requested discussions are needed at the end of results section.
Author Response
- The authors need to clarify what exactly the problem is and why this study is needed. This is
yet to be seen in the paper.
R:We added the lines 137-148, where is proposed that it does not exist studies in Chile about
housing prices with non-linear relationships and with intended to serve as an early warning.
Therefore, our work is pioneer to analyze the housing market with a more complex approach.
2. The novelty of the study must be highlighted in a detailed paragraph and compared with ex-
isting studies to show what innovation is brought in by this study. This can be added after
the last paragraph on page 1. Please note this is required in the introduction to justify why
this study is needed.
R: New lines inserted at lines 137-145 according your suggestion.
3. Please add a discussion section at the end of the result to compare the key ndings of this
study with existing works. The authors have provided some addition in conclusions but as
requested discussions are needed at the end of results section.
R: New lines inserted at lines 487-500 and 502-507 according your suggestion.
With kind regards.